# Development and Evaluation of a Physiologically Based Pharmacokinetic Model for Predicting Haloperidol Exposure in Healthy and Disease Populations

**DOI:** 10.3390/pharmaceutics14091795

**Published:** 2022-08-26

**Authors:** Mohammed S. Alasmari, Fawaz Alasmari, Abdullah F. Alasmari, Aws Alshamsan, Sary Alsanea, Muhammad F. Rasool, Faleh Alqahtani

**Affiliations:** 1Department of Pharmacology and Toxicology, College of Pharmacy, King Saud University, Riyadh 11451, Saudi Arabia; 2Department of Pharmaceutics, College of Pharmacy, King Saud University, Riyadh 11451, Saudi Arabia; 3Department of Pharmacy Practice, Faculty of Pharmacy, Bahauddin Zakariya University, Multan 60800, Pakistan

**Keywords:** PBPK, PK-Sim^®^, haloperidol, psychosis, personalized medicine

## Abstract

The physiologically based pharmacokinetic (PBPK) approach can be used to develop mathematical models for predicting the absorption, distribution, metabolism, and elimination (ADME) of administered drugs in virtual human populations. Haloperidol is a typical antipsychotic drug with a narrow therapeutic index and is commonly used in the management of several medical conditions, including psychotic disorders. Due to the large interindividual variability among patients taking haloperidol, it is very likely for them to experience either toxic or subtherapeutic effects. We intend to develop a haloperidol PBPK model for identifying the potential sources of pharmacokinetic (PK) variability after intravenous and oral administration by using the population-based simulator, PK-Sim. The model was initially developed and evaluated to predict the PK of haloperidol and its reduced metabolite in adult healthy population after intravenous and oral administration. After evaluating the developed PBPK model in healthy adults, it was used to predict haloperidol–rifampicin drug–drug interaction and was extended to tuberculosis patients. The model evaluation was performed using visual assessments, prediction error, and mean fold error of the ratio of the observed-to-predicted values of the PK parameters. The predicted PK values were in good agreement with the corresponding reported values. The effects of the pathophysiological changes and enzyme induction associated with tuberculosis and its treatment, respectively, on haloperidol PK, have been predicted precisely. For all clinical scenarios that were evaluated, the predicted values were within the acceptable two-fold error range.

## 1. Introduction

Schizophrenia is a very serious mental illness that usually results in disruption of a patient’s overall quality of life [1]. It has been reported that genetic, as well as environmental factors, get involved in the pathophysiology of this disease [2]. Based on the Diagnostic and Statistical Manual for Mental Disorders (DSM-5), a patient must satisfy certain criteria to be diagnosed with schizophrenia [3]. Even though the prevalence is relatively small, the presence of one schizophrenic patient in a family or a community costs a lot, economically and emotionally [4,5]. According to the World Health Organization (WHO), schizophrenia was ranked among the leading disorders contributing to the global disease burden [6].

Haloperidol is a typical, 1st generation, antipsychotic drug indicated primarily to treat psychological disorders, particularly, schizophrenia [7,8]. Its efficacy in psychosis and associated behaviors, including aggression and agitation, was demonstrated previously [7,8,9]. Recently, evidence has shown that haloperidol has a beneficial effect on substances-induced psychosis, nausea and vomiting associated with chemotherapy and palliative care, and delirium in intensive care units (ICU) [10,11]. Although, as a trend, the 2nd generation antipsychotics are preferred to be used over the 1st generation, studies found that haloperidol was either equally or more effective in improving mental health and delirium as compared to 2nd-generation antipsychotics, including risperidone, ziprasidone, quetiapine and aripiprazole in patients with acute schizophrenia [12,13]. Furthermore, many health institutions with limited resources still depend on haloperidol as a standard of care for several mental-related health issues.

It has been hypothesized that there is a strong correlation between psychosis and dopaminergic activity in the mesolimbic neuronal pathway of the brain. Neurotransmitter modulation by antipsychotic medications is thought to be the backbone intervention to relieve psychotic episodes in people with mental disorders [14,15]. The antagonizing effect on dopamine receptors by antipsychotics alleviated symptoms of psychosis. Pharmacologically, haloperidol exerts its effect by, non-selectively, blocking type-2 dopaminergic receptor (D_2_) on postsynaptic neurons in the central nervous system [16].

Due to some safety issues, it is not recommended to give haloperidol with doses above 30 mg daily [17], even though it is much lower than what has been mentioned as a maximum daily dose on the manufacturer’s label (100 mg/day). Moreover, although it is not uncommon for haloperidol to be administered intravenously in real-world clinical practice, this route of administration is not approved as per the US FDA haloperidol label [18]. Unpredictable serious adverse events including QTc prolongation and arrhythmia have been noticed to be correlated with the intravenous route, which could be explained partially by the inconsistent PK behaviors of haloperidol among individuals, causing a toxic biological exposure.

A large interindividual variability has been reported in human plasma concentration of haloperidol [19], which necessitates constructing a whole-body physiologically based pharmacokinetic (PBPK) model for better anticipation of its kinetic behavior [20]. Simultaneously, because of the narrow therapeutic window of haloperidol that warrants the application of precision medicine, the PBPK modeling has done much to advance the individualization of treatment according to a specific patient’s physiological needs [21]. In contrast to clinical PK experimentations that would be extremely costly to be conducted, the PBPK modeling can be used, alternatively, to predict a drug PK profile mathematically in virtually created human patient populations according to their characteristic features [20,22,23,24,25]. Moreover, pharmacological activities and toxic effects of xenobiotics can be predicted using the PBPK modeling by determining the target site exposure and affinity. In addition, the effects of drug interactions on biological exposure can be predicted using the PBPK modeling, which leads to proper dosing adjustment and avoids the unwanted effects of such interactions [26,27].

Haloperidol is extensively metabolized in the liver with about 1% of the administered dose being excreted renally in unchanged form. Hepatic glucuronidation, reduction, and cytochrome P450 (CYP)-mediated oxidation are the major metabolic biotransformation pathways for haloperidol [19,28]. In particular, drug disposition studies showed that uridine 5′-diphospho-glucanosyltransferases (UGT) 1A4, 1A9, 2B7, cytosolic carbonyl reductase-1 (CBR1), and CYP3A4, which are usually exposed to induction and inhibition, seem to be the major enzymes contributing to haloperidol metabolism [19,29]. Thus, the prediction of pharmacokinetic parameters as a result of such enzyme inhibition or induction using the PBPK model is highly recommended. In addition, reduced haloperidol (RHAL), which is the major byproduct of the reduction metabolic pathway, has the potential to be pharmacologically active or back oxidized to the active parent drug [30]. Therefore, it might be beneficial to kinetically characterize this metabolite and link it to haloperidol in a parent-metabolite PBPK model. Moreover, because haloperidol is one of the highly protein-bound drugs [31], its distribution might be affected significantly in pathological conditions where the level of plasma protein is influenced, such as tuberculosis (TB) [32]. Thus, we aim in this study to develop and evaluate whole-body parent-metabolite, drug-disease, and drug-interaction PBPK models for haloperidol to predict its exposure in the human population. In addition, the model will be used to suggest a dosing strategy to have a comparable therapeutic effect for TB patients who are using rifampicin (enzyme inducer) and haloperidol concomitantly.

## 2. Materials and Methods

### 2.1. Software

The software PK-Sim (version 9) which is applied by Bayer as part of the Open System Pharmacology (OSP) Suite (accessed via https://www.open-systems-pharmacology.org) was used as a platform to build the PBPK model for haloperidol in the virtual human population. It has been verified to accurately describe the effect of many internal and external factors that may influence the PK of medications in different medical scenarios that haven’t been evaluated clinically [33]. The software, mainly, consists of several building blocks that together make it possible to predict the PK profiles of drugs. Generally, the software comprises of different biological compartments that are connected through the arterial and venous blood flow [25,34]. It is a high-level software supported for modeling all PK-related processes including absorption, distribution, metabolism, and elimination (ADME) in one complete simulation while accounting for thorough biological and physiological representations. Moreover, PK-solver [35] was used for performing a non-compartmental analysis (NCA) on the observed and predicted data for calculating the PK parameters [35]. Additionally, the Get-Data Graph Digitizer^®^ program version 2.26 (accessed via http://getdata-graph-digitizer.com) was used to digitize the plasma concentration vs. time points from different reported studies, so that they can be overlaid on the predicted PK profiles [36].

### 2.2. Literature Search

A systematic literature search was conducted to gather the relevant information needed to build and evaluate the haloperidol PBPK model. Physicochemical properties of haloperidol and reduced haloperidol were obtained from different electronic databases including PubChem [37], DrugBank [38], and Human Metabolome Database (HMD) [39], while MEDLINE, EMBASE, and GOOGLE SCHOLAR were used to allocate the clinical trial-related information. In addition, references to related studies were checked for additional relevant PK studies to be considered for evaluation. The search strategy for clinical PK studies was limited to those that were written in English and conducted on humans. Only those clinical studies were included that contained systemic concentration vs. time curves, with clear information on drug administration protocol and type of population. The studies that were finally included are summarized in Table 1.

### 2.3. General Schematic Pathway

The previously reported practices were followed to develop the current PBPK model of haloperidol (Figure 1) [24,51,52,53]. Selected protocols describing PBPK modeling using PK-Sim as a platform have been identified to be studied and reviewed before developing the current model [54,55]. Generally, a collection of pharmacological, biological, physiological, and experimental data have been gathered together in one PK-Sim platform. A summary of input data used to develop the PBPK model of haloperidol is presented in Table 2. The model verification process was performed by comparing the predicted and observed values visually and numerically and, if necessary, model parameters were optimized for model fitting purposes. Then, the model was applied to predict the PK of haloperidol in rifampicin-naïve (control group) and rifampicin-experienced (intervention group) TB populations. Subsequent dosing regimens have been tested in those taking rifampicin to overcome the effect of interaction on the haloperidol exposure. Additionally, PBPK models for reduced haloperidol as a parent and as a metabolite have been developed and evaluated, separately, in the same manner. Drug-dependent properties for reduced haloperidol metabolite have been summarized in Table 3. A predefined template model for rifampicin was already built and incorporated in the PK-Sim and it has been used to induct the CYP3A4 enzyme and predict the effect of drug interaction with haloperidol.

### 2.4. Absorption, Distribution, Elimination, and Absorption (ADME)

The model was started with the simulation of haloperidol PK profile in an adult healthy population after intravenous administration for modeling the distribution and elimination phases. Drug-related properties such as fraction unbound (fu) and lipophilicity (logP) have been used as inputs for modeling the distribution. In the distribution phase, prediction methods that were used for partition coefficients and cellular permeability were those adopted by Rodger and Roland, and PK-Sim, respectively.

As haloperidol is one of the drugs that is almost completely metabolized with only about 1% of the administered dose excreted renally in the unchanged form [19], in vitro catalytic activities of enzymes were used as elimination determinants for modeling elimination. Functional activities, reference concentrations, and expression data of the targeted enzymes have been taken into account while developing the model. Haloperidol was found to be mainly metabolized by cytochrome P450 3A4 (CYP3A4), carbonyl reductase (CBR1), and UDP-glucuronosyltransferase (UGT) 1A4, 1A9, 2B7 [19,28,29]. Enzymatic biotransformation of haloperidol has been described in Table 2.

For CBR1, the catalytic activity has been calculated based on in vitro Km and Vmax from human liver cytosol (HLC) [19,56]. The abundance of CBR1 protein has been reported previously in several studies, with a wide range of variability [57]. The content of CBR1 protein in human liver cytosol is 75 [pmol/mg cytosolic protein] [58], which is corresponding to a reference enzyme concentration of 5.6 µmol/L resulting from the multiplication of 75 [pmol/mg cytosolic protein] by 75.4 [mg cytosolic protein/g liver] [59], and then divided by 1000, accounting for liver density.

Oxidation of haloperidol to pyridinium metabolite (HPP^+^) is catalyzed solely by CYP3A4. Using a microsomal extract from several human hepatic tissues, it has been shown that pyridinium metabolite was formed with Km and Vmax of 80 µmol/L and 0.53 pmol/min/pmol, respectively [19,60]. Before implementing the value of Vmax in the software, the unit has been converted to pmol/min/mg by multiplying 0.53 [pmol/min/pmol] by 460 [pmol/mg] (microsomal CYP content) [19]. Thus, the Vmax value for the transformation of haloperidol to its pyridinium metabolite was 243.8 pmol/min/mg.

In addition, CYP3A4 was found to catalyze another oxidation reaction for haloperidol, which is N-dealkylation, leading to the formation of para-fluorobenzyl-propionic acid (4-FBPA). A wide variability (up to 10-fold) between individuals has been shown in terms of the catalyzation of N-dealkylation [19].

Glucuronidation is the major metabolic pathway through which more than 50% of haloperidol is eliminated. Haloperidol O-glucuronide (major) and N-glucuronide (minor) are two glucuronides that have been identified experimentally [29]. UGTs enzymes that are responsible for catalyzing this pathway have been clarified in terms of enzyme isoforms that are involved in the process (1A4, 1A9, and 2B7), and their kinetic activities. As that recombinant enzymatic system has been used to study kinetic characteristics for each isoform, the Intersystem Extrapolating Factor (ISEF) method has been used to scale the data from recombinant system to human liver microsome (HLM) as described previously [61]. In addition, apparent Km values were corrected for non-specific binding resulting from incubational assays.

The absorption phase of oral haloperidol was developed and verified, subsequently. Specific intestinal permeability (P_int_) has been calculated directly from the effective molecular weight (EMW), lipophilicity, and membrane affinity (MA) [62,63]. After modeling the distribution and elimination phases based on intravenous administration, modeling the absorption was performed from oral administration, with parameters such as drug solubility and intestinal permeability being incorporated in the model. In PK-Sim, the compartmental gastro-intestinal model is used to describe the absorption phase. To have a good agreement between predicted and observed data, parameter identification and manual optimization have been applied to some values according to reported ranges.

**Table 2 pharmaceutics-14-01795-t002:** Input parameters used to build the haloperidol PBPK model.

Parameter	Unit	Input Value	Reported Value(s) (Reference)	Source of Input
**Physicochemical Properties**				
Molecular weight	g/mol	376	376 [39]	[39]
Effective molecular weight	g/mol	336.86		PK-Sim
Lipophilicity	Log	3.66	2.9 [64], 3.01 [28], 3.20 [37], 3.23 [65], 3.66 [38], 3.7 [38], 4 [39], 4.3 [37]	[38]
Water solubility	mg/mL	0.0045	0.0045 [38,39], 0.014 [37], 0.023 [37]	[39]
pKa-monoprotic base		8.65	7.8 [65], 8.65 [66], 8.05 [38,39]	[66]
**Absorption**				
Intestinal permeability	cm/min	3.09 × 10^−4^	14.5 × 10^−6^ (cm/s) [28]	Calculated
**Distribution**				
Fraction unbound	%	8.5	7.5–11.6 [19], 12.5 [65], 16.6 [28]	Parameter identification
Partition coefficient model		Rodger and Roland		PK-Sim
Cellular permeability model		PK-Sim		PK-Sim
**Renal Clearance**				
GFR fraction	%	1	1	[19]
**Enzymatic Biotransformation**				
Enzyme: CYP3A4 Metabolite: dealkylation to 4-FBPA In vitro metabolic system: Human liver microsome				
K_m_ *	µmol/L	21.7	62 [19]	[19]
V_max_	pmol/min/mg	289	289 [19]	[19]
Enzyme: CYP3A4 Metabolite: oxidation to HPP^+^ In vitro metabolic system: Human liver microsome				
K_m_ *	µmol/L	28	80 [19]	[19]
V_max_	pmol/min/mg	243.8	0.53 pmol/min/pmol [19]	Calculated based on 460 pmol/mg [19]
Enzyme: Carbonyl Reductase Metabolite: cytosolic reduction to reduced haloperidol In vitro metabolic system: Human liver cytosol				
K_m_ *	µmol/L	88	250 [19]	[19]
V_max_	nmol/min/mg	1.30	1.30 [19]	[19]
Reductase content in liver cytosol	pmol/mg	75	75 [58]	[58]
Enzyme: UGT1A4 Metabolite: O-glucuronidation In vitro metabolic system: Recombinant system				
K_m_ *	µmol/L	22.4	64 [29]	[29]
V_max_	pmol/min/mg	600 × ISEF_UGT1A4_	600 [29]	[29]
Content in liver microsome	pmol/mg	33	33 [67]	[67]
ISEF_UGT1A4_		0.16		Calculated
Enzyme: UGT1A9 Metabolite: O-glucuronidation In vitro metabolic system: Recombinant system				
K_m_ *	µmol/L	61	174	[29]
V_max_	pmol/min/mg	2300 × ISEF_UGT1A9_	2300 [29]	[29]
Content in liver microsome	pmol/mg	22.7	22.7 [67]	[67]
ISEF_UGT1A9_		0.06		Calculated
Enzyme: UGT2B7 Metabolite: O-glucuronidation In vitro metabolic system: Recombinant system				
K_m_ *	µmol/L	16	45.0	[29]
V_max_	pmol/min/mg	1000 × ISEF_UGT2B7_	1000 [29]	[29]
Content in liver microsome	pmol/mg	69.4	69.4 [67]	[67]
ISEF_UGT2B7_		0.05		Calculated
Enzyme: UGT1A4 Metabolite: N-glucuronidation In vitro metabolic system: Recombinant system				
K_m_ *	µmol/L	64	64	
V_max_	pmol/min/mg	440 × ISEF_UGT1A4_	210 [29]	Calculated based on reported V [29]
Incubational fraction unbound		0.35	0.34 [31], 0.35 [64], 0.75 [28]	[64]

* Apparent K_m_ values used were corrected for the non-specific binding (fu.inc) from in vitro incubational assays. ISEF = intersystem extrapolating factor used to scale data obtained with recombinant system to liver microsomal system. V = reaction velocity. GFR = glomerular filtration rate. 4-FBPA = 4-Fluorobenzoylpropionic acid. HPP+ = Haloperidol pyridinium derivative. HLM = human liver microsome.

**Table 3 pharmaceutics-14-01795-t003:** Input data for the PBPK model of reduced haloperidol metabolite.

Parameter	Value	Source
**Physicochemical Properties**		
Molecular weight	378	Drug-Bank
Effective molecular weight	339	
LogP	3.52	
pKa	8.66	
Solubility	0.0131 mg/mL	
**Absorption**		
Intestinal permeability	2.08 × 10^−4^ (cm/min)	Calculated
**Distribution**		
Partition coefficient model	Rodger and Roland	PK-Sim
Cellular permeability model	PK-Sim	PK-Sim
Fraction unbound	24.4%	[68]
**Elimination**		
Enzyme: CYP3A4 Metabolic pathway: oxidation to haloperidol In vitro metabolic system: Human liver microsome		
K_m_	46 μmol/L	[19]
V_max_	98 pmol/min/mg	[19]
Additional hepatic clearance	5 mL/min/kg	Estimated

### 2.5. The Predictability Assessment

The evaluation strategy was performed as per guidelines for the evaluation of PBPK models [24]. Initially, time versus concentration profiles was appraised visually based on the general pattern of graphical representations of predictions with experimental observations. The model is considered to be visually verified if a reasonable agreement has been reached between the simulated and experimental PK data, considering that experimental observations should be within the 5th and 95th percentiles of the predictive data. To numerically evaluate the model performance, prediction error (PE%), mean fold error (MFE), and mean square root of error (RMSE) were used as indicators for determining the extent to which the predicted values have deviated from the observed values (Equations (1)–(4)) [69,70,71].
(1)$$\mathrm{Ratio}=\frac{Predicted\;value\;of\;PK\;parameter\;}{Observed\;value\;of\;PK\;parameter\;}$$
(2)$$MFE=\frac{Mean\;of\;prediced\;values\;}{Mean\;of\;observed\;values\;}$$
(3)$$\;PE\;\left(\%\right)=\frac{Prediced\;value-Observed\;value\;}{Observed\;value}\times 100$$
(4)$$RMSE=\sqrt{\frac{\sum_{1}^{N}{\left(observed\;value-predicted\;value\right)}^{2}}{N}}$$

The acceptable limits for PE and MFE have been determined to be in the 25% and 2-fold range, respectively, as suggested previously [69,70,71]. The box-whisker plots were used to display the effect of drug interactions and dosing suggestions. For estimating the effect of drug interaction on haloperidol exposure, concentration–time profiles of haloperidol with and without the perpetrator drug, which is rifampicin 600 mg orally, have been compared graphically. In addition, the ratio of predicted area under the curve extrapolated to infinity (AUC^0–∞^), the concentration at the end of the simulation (C__tEnd_), and half-life (T_1/2_) were compared according to Equations (5)–(7).
(5)$$\;AUC\;ratio=\frac{AUC\;for\;CYP3A4\;induction\;}{AUC\;for\;control}$$
(6)$$C_{\_tEnd}\;ratio=\frac{C_{\_tEnd}\;for\;CYP3A4\;induction\;}{C_{\_tEnd}\;for\;control}$$
(7)$$T1/2\;ratio=\frac{T1/2\;for\;CYP3A4\;induction\;}{T1/2\;for\;control\;}$$

### 2.6. PK of Haloperidol in Disease Population

Psychosis does not significantly impact haloperidol pharmacokinetic properties [46,72]. The reported unbound fractions of haloperidol are comparable in healthy and psychotic populations [19]. Thus, we assume that no physiological parameters needed to be adjusted for the psychotic, otherwise healthy, population. On the other hand, a quiet alteration in the distribution behavior of haloperidol has been noticed in TB patients [45]. This can be attributed to the effect of TB on the contents of plasma proteins including albumin and globulin as described previously [32], which has the potential to affect the free fraction of the drugs. To account for this physiological status, a plasma protein scaling factor has been used to estimate the effect of TB on the free fraction of haloperidol as recommended (PK-Sim 7.0.0 manual—page 85). In comparison to anti-TB treatment naïve patients, a significant reduction in the haloperidol level has been shown in anti-TB treatment-experienced patients, which was mainly attributed to rifampicin as a potent CYP3A4-inducer [45].

## 3. Results

### 3.1. Clinical PK Studies That Were Used to Build and Evaluate the Haloperidol PBPK Model

Haloperidol in dosage ranges of 2.5–10 mg intravenous, and 2–35 mg oral, was evaluated kinetically in human populations through several PK studies (Table 1). Out of these studies, two were conducted on healthy volunteers with the same population demographics [40,41]. In both studies, PK analysis was performed for haloperidol being administered orally and intravenously, and the concentration was determined in the serum using the same analyzing techniques. Another study was conducted on an Asian, schizophrenic, otherwise healthy population with a similar experimental design and dosing protocols to the previously mentioned studies [42]. Experimentally, higher plasma concentration for haloperidol was noticed in the Asian population in comparison to the non-Asian. In a single-dose study [49], 5 mg oral haloperidol was characterized kinetically in 36 healthy individuals, and it has been shown that there is a great deal of interindividual variability in PK parameters. In another study, pharmacokinetic analysis of intravenous haloperidol was performed based on 10 healthy volunteers and 2 psychotics, otherwise healthy. [43]. The study was included because of the availability of the relevant information for model evaluation such as the dose and concentration–time profile.

On the other hand, two studies evaluated the PK properties of intravenously [44,46], and orally [46] administered haloperidol in psychotic, otherwise healthy, individuals. Cheng and his colleagues (1987) studied haloperidol PK profile in psychotic males and females between 20 and 43 years of age and used the HPLC technique to detect the serum concentration. Generally, they found that pharmacokinetic parameters were comparable to those reported according to healthy volunteers. In another study, haloperidol was characterized and assessed kinetically in the psychotic tuberculotic population [45]. Plasma concentration–time profile for 5 mg intravenous haloperidol with and without antituberculosis agent, rifampicin, was studied. The study was included to evaluate the effect of TB on the PK of haloperidol and test the ability of the model to predict the effect of drug interaction and suggest dosing regimens in the presence of concomitant enzyme inducer. In addition, a separate PBPK model for reduced haloperidol metabolite was developed and evaluated subsequently with respect to observed data from two clinical PK studies [42,48].

### 3.2. The PBPK Model for Adult Healthy and Psychotic, Otherwise Healthy, Populations after Intravenous and Oral Administration

The developed haloperidol PBPK model was able to precisely describe the concentration in the plasma versus time profiles in adult healthy populations. Figure 2 and Figure 3 illustrate the evaluated intravenous and oral haloperidol PBPK model for the virtual population of 100 healthy individuals. The model’s validity has been verified visually (Figure 2, Figure 3 and Figure 4), and by the comparison of predicted-to-observed individual PK data profiles, as presented in Table 4 and Table 5. All simulated values were within the predefined two-fold range of the experimental data. In the case of haloperidol being administered intravenously, MFE of the model values for the area under the curve (AUC__inf_), maximum concentration (C__max_), and clearance (CL) were 0.91, 1.0, and 1.02, respectively. For the model of oral haloperidol, MFE for AUC__inf_, C__max_, and CL were 0.99, 1.03, and 0.99, respectively. In addition, the performance of the models was validated according to percentages of PE which are lower than the predefined acceptable limit, as presented in Table 4 and Table 5.

### 3.3. Reduced Haloperidol as a Parent and as a Metabolite

PBPK models for reduced haloperidol as a parent and as a metabolite from haloperidol were developed and evaluated. The model was able to precisely capture the observed data very well. As shown in Figure 5A, observed clinical data of 10 mg oral reduced haloperidol were predicted to be within the 5th to 95th prediction range. Moreover, the model successfully predicted the PK behavior of reduced haloperidol as a metabolite after oral and intravenous administration of haloperidol (Figure 5B–D) with most of the data points being within the predefined prediction range.

### 3.4. The PBPK Model for Tuberculotic Population and Evaluation of Its Performance to Predict Drug Interaction with the Anti-Tuberculotic Agent and Dose Optimization

The validity of the haloperidol PBPK model for the control group of TB population (anti-TB treatment naïve group) has been qualified according to visual predictive checks given that all of the observed data were within the 5th to 95th percentile range (Figure 6). A good agreement has been reached between predicted and observed data [61] (mean of two TB patients taking no anti-TB drugs) as evidenced by the ratio and the mean fold error, as illustrated in Table 6. MFE and PE were found to be within the acceptable predefined limits of error.

The effect of rifampicin as a CYP3A4 enzyme inducer on the plasma concentration of haloperidol was precisely predicted by the PBPK modeling and it has been presented in Figure 7. In addition, Box whisker blots have been used to evaluate the effect of drug interaction on PK parameters of haloperidol, as shown in Figure 8. The model predictability was able to capture the clinical observed data very well from the plasma profile of psychotic tuberculotic patients taking haloperidol and rifampicin concomitantly, as illustrated in Table 6. In comparison to haloperidol alone (control group), about 50% decrease in the AUC and half-life of haloperidol has been noticed in those treated with rifampicin, which is in line with what has been noticed clinically from the observed data [61]. As shown in Table 6, the corresponding ratio of predicted AUC and half-life between control and rifampicin groups were 0.6 and 0.4, respectively, indicating that there is a roughly 50% decrease in the exposure of haloperidol as a result of interaction with rifampicin.

To have a comparable haloperidol exposure in the intervention group (combination group), we have tried to increase the dose by 50%, 75%, and 100% (Figure 9). It seems that a 75% increase in the dose of haloperidol is appropriate to have a comparable pharmacological exposure to the control group (haloperidol alone). The means of total AUC in the control and intervention groups subjected to a 75% increase in haloperidol dose are 192.3 and 194.3 ng·h/mL, respectively.

## 4. Discussion

In this study, the whole-body PBPK models of haloperidol and reduced haloperidol, for healthy adults and psychotic, otherwise healthy, populations, have been successfully developed and evaluated with all predictions being within the predefined range of error. The haloperidol PBPK model was able to precisely describe oral and intravenous plasma concentration–time profiles and predicts the PK parameters in the populations of interest. In addition to its applicability in healthy and psychotic, otherwise healthy, populations, it was demonstrated that the haloperidol PBPK model can be applied to predict PK parameters in TB patients and the effect of interaction with anti-TB rifampicin, which is a potent CYP3A4 inducer. The functionality of the PBPK model was verified graphically and numerically by creating population PK profiles and by the comparison of the simulated and the clinical experimental data. Results from MFE and PE for PK parameters were calculated, and the performance of the model was statistically demonstrated. Specifically, MFE for all simulations was within a two-fold range of the observed data and the PE percentages were lower than the predefined acceptable limits.

The effective dose of haloperidol in patients with acute change in cognitive function is generally 0.5 to 20 mg either IV or PO, depending on care settings. It is recommended to start with lower doses and gradually titrate up to a maximum of 30 mg daily, according to the therapeutic response. In the comparison of studies that followed the weight-based dosing strategy (0.125 mg/kg IV) [40,41] to the study that used a fixed-dose protocol of 10 mg IV [43], the plasma concentration profile appears to be comparable, as shown in (Figure 4A), with mean AUC from time zero to infinity of 211.38 ng/mL/h. Administration of 0.5 mg/kg orally (a total dose of more than 30 mg based on the mean weight of the population) results in high drug exposure (around 500 ng/mL/h). This illustrates the reason why it is not recommended to go above 30 mg daily, which may increase the risk of the adverse event. The developed PBPK model confirms that the PK of haloperidol in healthy and psychotic populations are comparable. For graphical evaluation, we combined two or more observed data based on the route and dosage range as recommended clinically, since intravenous haloperidol has to be started at lower doses and go slow to avoid unwanted adverse events. All observed data were within the 5th to 95th prediction range.

Haloperidol is one of the most commonly used drugs that is prescribed for several label and off-label indications. Despite the comparability of PK parameters’ means of the haloperidol among different patient populations, large between-individual variability of the drug exposure has been noticed. Previous attempts, based on population pharmacokinetic approaches, have been made to explain the variations of the haloperidol PK parameters between individuals, regardless of population type [74,75,76]. Studies were conducted using non-linear mixed effects modeling in order to give a real insight into the causes of the variability and to help with an individualized treatment approach. It was found that kinetic behaviors of haloperidol were adequately described by one- and two-compartment models. No explanations were found for the inherent variability between individuals with the covariates analyses that have been used [74]. In particular, no clear correlation has been noticed in term of some covariates, including body weight and plasma level of bilirubin, and the PK parameters. Moreover, body weight was not found to be correlated with haloperidol clearance according to a recently published population pharmacokinetic analysis, while a negative correlation has been established with C-reactive protein [76]. On the other hand, a slight correlation has been found between body weight and PK parameters, but this was based on a Japanese population, which might be attributed to the inherent differences related to ethnicity [77]. Generally, it is well documented that body weight is an important factor that has an impact on clearance [78], however, in terms of haloperidol, the unexplained variation in plasma concentration was present even in a population with comparable baseline characteristics.

The sources of interindividual variability are widely varied and can be either related to anatomy or physiology [79]. In most cases, it has a huge impact on observed plasma concentrations, which may increase the risk of either toxic or subtherapeutic levels of a candidate drug in a subgroup of individuals in comparison to population averages [80]. PBPK modeling is one of the right tools that is being used to explain the raised issues related to the variability between individuals in terms of PK parameters [79,80,81]. One study found that in addition to individual variations in tissue volume and their related blood flow, the plasma concentrations are largely influenced by the variations in partition coefficients between individuals [79]. Moreover, some variability with unknown sources was attributed to tissue compositions. In another study, the PK-Sim was used to evaluate the interindividual variability on the PK parameters of ciprofloxacin and paclitaxel [81]. They made use of information about means and distributions of the anthropometric properties based on what has been published in the literature and found that considering interindividual variations lead to well-predicted PK profiles. Thus, it would be helpful to consider such variations while predicting PBPK-based PK profiles. Due to the unexplained wide interindividual variability that is linked to haloperidol pharmacokinetics, it was necessary to perform a PBPK model with physiological and anthropometric properties and their distributions in the real population being implemented. To the best of our knowledge, this study is the 1st comprehensive study that evaluates haloperidol kinetically using the PBPK approach. The capability of our haloperidol PBPK model for prediction was high enough to capture the observed data successfully. This, partially, was attributed to the nature of the PBPK tool that takes several drug/biological systems-related components into account while predicting the PK profiles. In our study, the predictability of the model was increased obviously, with the addition of intrinsic clearance variability based on the reported values from the literature. Creating a virtual population based on a virtual individual considering distribution of covariates among individuals improves our population model performance.

In addition to the previously mentioned factors that may contribute to the observed variability in haloperidol PK, we assume that the unpredictability in the haloperidol plasma concentration can be mainly attributed to the wide variations in the catalytic activities of the whole system responsible for the enzymatic biotransformation. Since that haloperidol is extensively metabolized in humans, a large interindividual variability in its pharmacokinetics is expected [19]. From the simulation perspective, the most determinant parameter for the haloperidol PBPK model to best fit the observed data is the ability to model the metabolic biotransformation very well. Concentrations and functional activities of the enzymes significantly impact the overall performance of the model to properly predict the drug exposure. In comparison to a previous PBPK model for IV haloperidol [31], which was evaluated based on a single clinical PK study [46], our study was able to describe the elimination phase more precisely. Partially, this can be attributed to the metabolization inputs that were incorporated to build the model, in comparison to what has been incorporated in the previous study. One of the advantages of this study is the inclusion of reduced haloperidol metabolite in the model, which is the major metabolite. Haloperidol was detected in the blood after pure reduced haloperidol was given to an individual [48]. This indicates the importance of the incorporation of metabolic activity of haloperidol in the PBPK modeling. On the other hand, despite the availability of other administration routes that have been tested, such as intramuscular and intranasal [47], only oral and intravenous administration routes have been explained by this model; however, modeling other routes of administration can be performed in future studies. In addition, since only enzyme induction has been tested, the effect of CYP3A4 inhibition also needs to be studied.

## 5. Conclusions

The current haloperidol PBPK model has successfully described the kinetic behavior of haloperidol after oral and intravenous administration in all medical scenarios that were evaluated. The functionality of the model to be applied in healthy/psychotic, otherwise healthy, and TB populations was verified visually and numerically. The first PBPK model for the fating of haloperidol to its reduced metabolite and vice versa has been established. The model showed a significant drug interaction through the CYP3A4 enzyme, resulting in an approximately 50% reduction in haloperidol exposure. A subsequent increase in the dose was needed to have comparable PK parameters to the control situation. Clinically, this can be meaningful and should be considered for those who are using multiple chronic medications. The large interindividual variability associated with haloperidol exposure can be attributed to the wide range of variations in the catalytic activities of enzymes among individuals. The involvement of several metabolizing enzyme isoforms with different activities and tissue contents can be one of the main sources of the observed variability in haloperidol biological exposure.

## Figures and Tables

**Figure 1 pharmaceutics-14-01795-f001:**
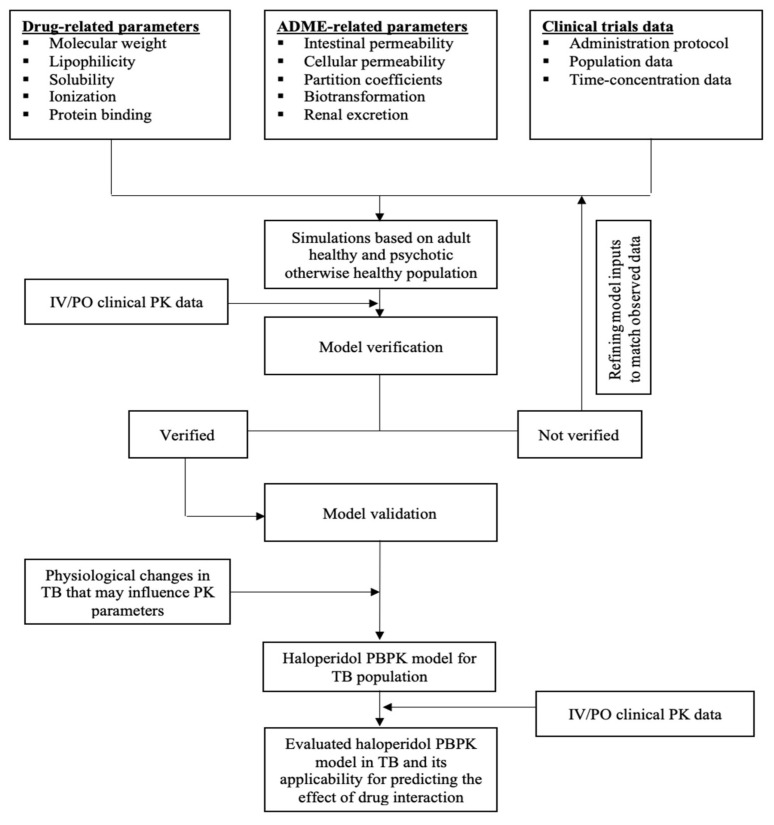
Workflow for the development of haloperidol PBPK model.

**Figure 2 pharmaceutics-14-01795-f002:**
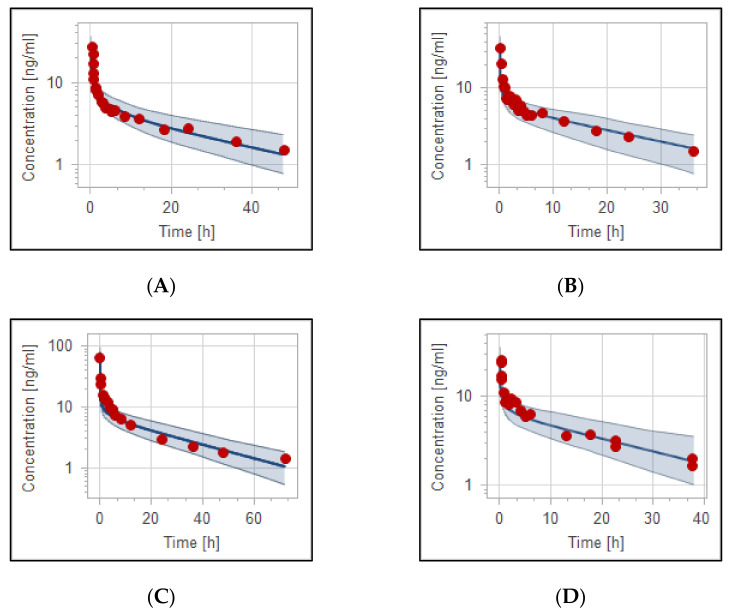
Plasma concentration–time profiles of haloperidol in adult healthy and psychotic otherwise healthy populations after intravenous administration. (**A**) 0.125 mg/kg [40], (**B**) 0.125 mg/kg [41], (**C**) 0.125 mg/kg IV [42], (**D**) 10 mg IV [43], (**E**) 5 mg IV [44], (**F**) 3.5 mg IV [46], (**G**) 2.5 mg IV [47]. Observed data are depicted as red circles and means of simulated data are depicted as solid blue lines with the 5th–95th predictive ranges as blue shaded area.

**Figure 3 pharmaceutics-14-01795-f003:**
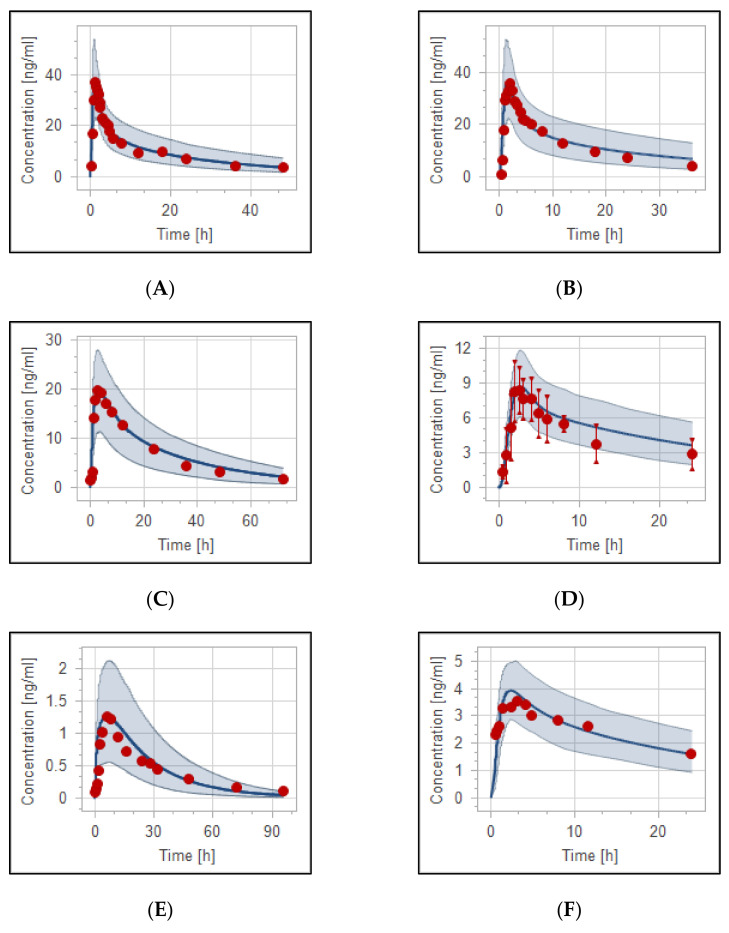
Plasma concentration–time profiles of haloperidol in adult healthy and psychotic otherwise healthy populations after oral administration of haloperidol. (**A**) 0.503 mg/kg [40], (**B**) 0.5 mg/kg [41], (**C**) 0.5 mg/kg [42], (**D**) 10 mg [48], (**E**) 5 mg [49], (**F**) 2 mg [46]. Observed data are depicted as red circles and means of simulations are depicted as solid blue lines with the 5th–95th predictive range as blue shaded area.

**Figure 4 pharmaceutics-14-01795-f004:**
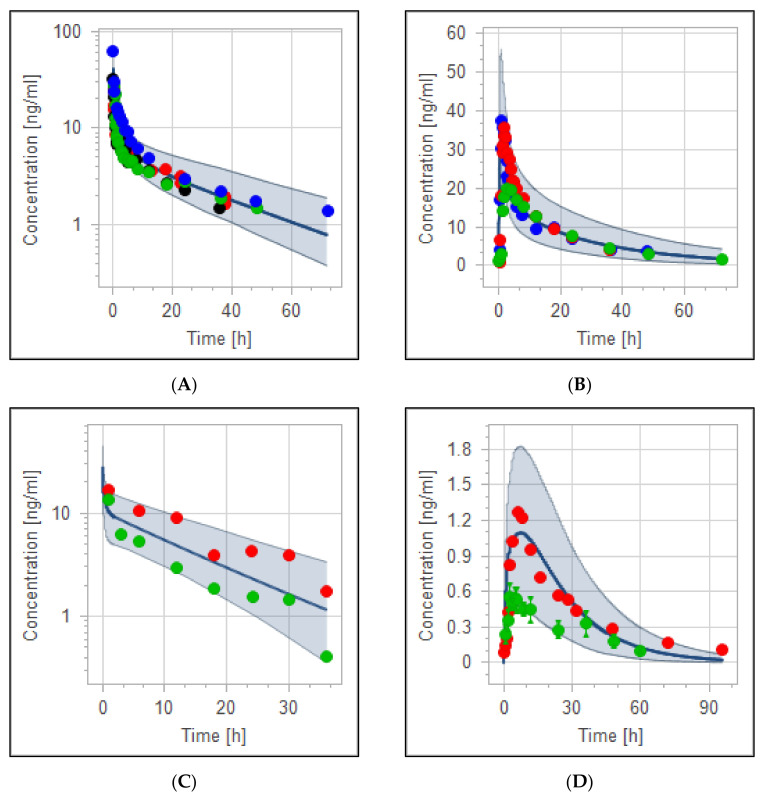
Visual verification of haloperidol population PBPK model. Observed data have been combined into one simulation according to the similarity in the dosage regimen and administration protocol. (**A**) PK profile of healthy and psychotic otherwise healthy populations after IV administration with experimental data depicted as blue [42], green [40], red [43] and black [41] circles. (**B**) PK profile of adult healthy population after oral administration with experimental data being depicted as red [41], blue [40], and green [42] circles. (**C**) PK profiles of adult TB population (treatment-naïve) after haloperidol 5 mg IV with experimental data depicted as red and green circles [45]. (**D**) Adult healthy/psychotic otherwise healthy population after 5 mg oral haloperidol with observed data depicted as red [49] and green [73] circles. Blue solid lines and shaded areas illustrate simulation mean and the 2.5th to 97.5th prediction range, respectively.

**Figure 5 pharmaceutics-14-01795-f005:**
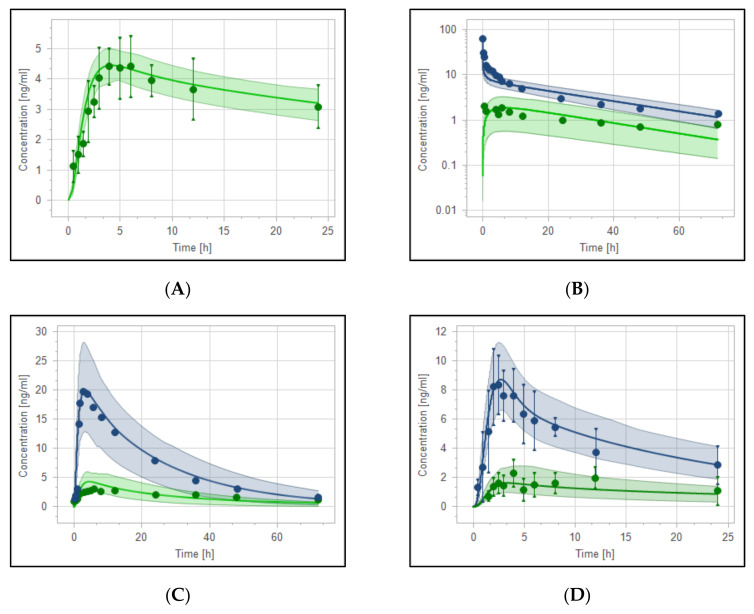
Plasma-concentration time curves of haloperidol (blue) and reduced haloperidol metabolite (green). (**A**) Simulated and observed data of 10 mg reduced haloperidol administered orally [48]. (**B**–**D**) Haloperidol (blue) and reduced haloperidol (green) after 0.125 mg/kg IV [42], 0.5 mg/kg oral [42], and 10 mg oral haloperidol [48], respectively. Observed data are depicted as circles and means of simulations are depicted as solid lines with the 5th–95th predictive range being depicted as shaded areas.

**Figure 6 pharmaceutics-14-01795-f006:**
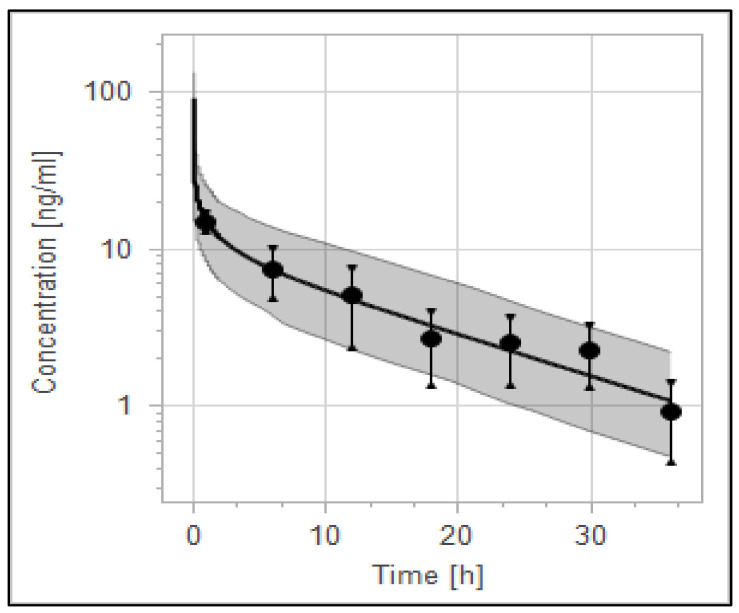
Concentration–time profile of haloperidol in TB patient.

**Figure 7 pharmaceutics-14-01795-f007:**
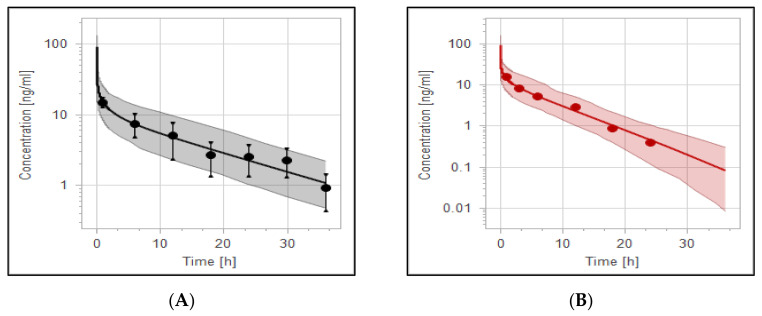
Effect of tuberculosis on PK of haloperidol with and without anti-TB rifampicin as a CYP3A4 inducer. (**A**) Haloperidol 5 mg IV in TB patients not treated with rifampicin. (**B**) Haloperidol 5 mg IV treated with rifampicin. (**C**) Simulation comparison for haloperidol with and without rifampicin. The red solid line represents the mean plasma concentration of haloperidol alone, while the black solid line represents the mean plasma concentration of haloperidol plus rifampicin. Colorful circles are observed data for haloperidol with (black) and without (red) rifampicin.

**Figure 8 pharmaceutics-14-01795-f008:**
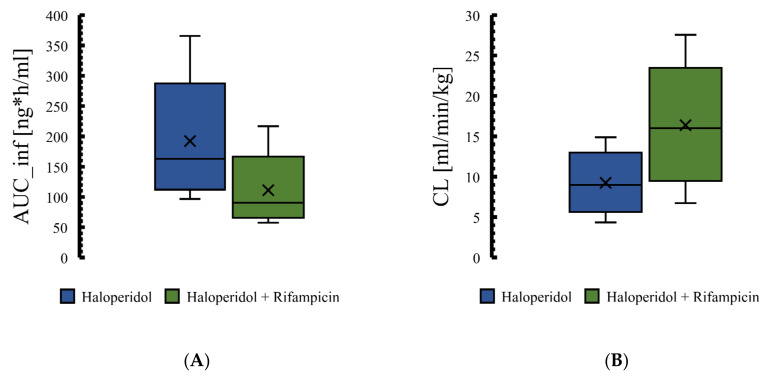
Plasma Box-whisker plots comparing pharmacokinetic parameters of haloperidol as a result of interaction with rifampicin. Figures (**A**–**D**) illustrate the effect of CYP3A4 induction by rifampicin on the haloperidol pharmacokinetic parameters. AUC: area under the curve, CL: clearance, C__tEnd_: concentration at the end of the simulation.

**Figure 9 pharmaceutics-14-01795-f009:**
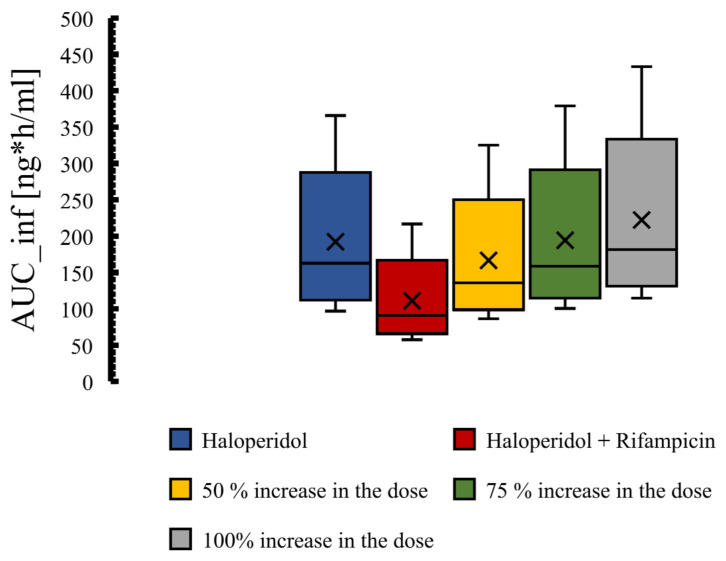
Dosing optimization to overcome the effect of PK interaction between haloperidol and rifampicin. 50%, 75%, and 100% increase in the dose of haloperidol in the combination group.

**Table 1 pharmaceutics-14-01795-t001:** List of clinical pharmacokinetic studies used for developing and evaluating the haloperidol PBPK model.

Dose	N	Female (*n*)	Age (Year)	Weight (kg)	Population	Reference
**Haloperidol Administered Intravenously**						
0.125 mg/kg	6	0	21–37	Mean: 70.5	Healthy	[40]
0.125 mg/kg	12	0	19–37	56–92	Healthy	[41]
0.125 mg/kg	8	0	21–48	Mean: 63.3	Psychotic otherwise healthy	[42]
10 mg	12	0			Healthy	[43]
5 mg	1	1	Mean: 55	74	Psychotic otherwise healthy	[44]
5 mg	2		20 s–60 s		Tuberculotic (control)	[45]
5 mg	3		20 s–60 s		Tuberculotic (intervention)	[45]
3.5 mg	6	3	20–43	Mean: 67	Psychotic otherwise healthy	[46]
2.5 mg	4	2	24–37	63–82	Healthy	[47]
**Reduced Haloperidol after Haloperidol Administered Intravenously**						
0.125 mg/kg	8	0	21–48	Mean: 63.3	Psychotic otherwise healthy	[42]
**Haloperidol Administered Orally**						
0.503 mg/kg	8	0	19–37	Mean: 70.8	Healthy	[40]
0.500 mg/kg	9	0	19–37	56–92	Healthy	[41]
0.500 mg/kg	6	0	21–48	Mean: 63.3	Psychotic otherwise healthy	[42]
10 mg	6	0	32–57	43–66	Psychotic otherwise healthy	[48]
5 mg	28	0	18–50	Mean: 71.5	Healthy	[49,50]
2 mg	8	5	Mean: 32	Mean: 67	Psychotic otherwise healthy	[46]
**Reduced Haloperidol after Haloperidol Administered Orally**						
0.500 mg/kg	6	0	21–48	63.3 ± 6.7	Psychotic otherwise healthy	[42]
**Reduced haloperidol administered orally**						
10 mg	6	0	32–57	43–66	Psychotic otherwise healthy	[48]

**Table 4 pharmaceutics-14-01795-t004:** Pharmacokinetic analysis of predicted and observed values in adult healthy and psychotic otherwise healthy populations after intravenous administration.

PK Parameter	Predicted	Observed	Pre/Obs Ratio	PE (%)
**0.125 mg/kg IV in adult healthy** [40]				
AUC_inf [ng·h/mL]	168	202	0.83	17%
C_max [ng/mL]	28	27 *	1.04	4%
CL [mL/min/kg]	12.3	11.1	1.11	9.3%
**0.125 mg/kg IV in adult healthy** [41]				
AUC_inf [ng·h/mL]	171.3	173.3 *	1.01	1.2%
C_max [ng/mL]	32.5	32.5 *	1.00	0.00%
CL [mL/min/kg]	12.0	12.02 *	0.99	0.00%
**0.125 mg/kg IV in adult schizophrenic otherwise healthy** [42]				
AUC_inf [ng·h/mL]	271	383	0.71	6.3%
C_max [ng/mL]	77.22	63.10 *	1.22	22.4%
CL [mL/min/kg]	7.56	6.20	1.22	21.9%
**10 mg IV in adult healthy** [43]				
AUC_inf [ng·h/mL]	193	202 *	0.96	4.5%
C_max [ng/mL]	31.50	25.70 *	1.21	23%
CL [mL/min/kg]	12.20	11.90 *	1.03	2.5%
**5 mg IV in adult psychotic otherwise healthy** [44]				
AUC_inf [ng·h/mL]	97.56	86.20 *	1.13	13.2%
C_max [ng/mL]	108.70	115 *	0.95	5.5%
CL [mL/min/kg]	10	13 *	0.77	23%
**3.5 mg IV in adult psychotic otherwise healthy** [46]				
AUC_inf [ng·h/mL]	144.30	175.72 *	0.82	18%
C_max [ng/mL]	15.13	19.2 *	0.79	21.2%
CL [mL/min/kg]	6.30	6.50	0.97	3.1%
**2.5 mg IV in adult healthy** [47]				
AUC_inf [ng·h/mL]	56	60.40	0.93	7.3%
C_max [ng/mL]	22.00	27.43 *	0.80	19.8%
CL [mL/min/kg]	10.62	11.55 *	0.92	8.2%
**Predictability assessment**				
	**AUC_inf**	**C_max**	**CL**	
**MFE**	0.91	1.00	1.01	
**RMSE**	46.2	6.76	1.38	

* Observed PK values were calculated using the PK-solver, otherwise, the reported values in the corresponding references were used.

**Table 5 pharmaceutics-14-01795-t005:** Pharmacokinetic analysis of simulated and observed values after oral administration of haloperidol in healthy and psychotic otherwise healthy populations.

PK Parameter	Predicted	Observed	Pre/Obs Ratio	PE (%)
**0.503 mg/kg Oral in Adult Healthy** [40]				
AUC_inf [ng·h/mL]	537.4	566	0.95	5.1%
C_max [ng/mL]	33.60	37.4 *	0.90	10.2%
CL [mL/min/kg]	15.61	15.41 *	1.01	1.3%
**0.500 mg/kg Oral in Adult Healthy** [41]				
AUC_inf [ng·h/mL]	544	500.1 *	1.10	8.8%
C_max [ng/mL]	35.34	35.83 *	0.99	1.3%
CL [mL/min/kg]	15.20	16.64 *	0.91	8.7%
**0.500 mg/kg Oral in Adult Schizophrenic Otherwise Healthy** [42]				
AUC_inf [ng·h/mL]	550	512.6	1.07	7.3%
C_max [ng/mL]	20.81	19.76 *	1.05	5.3%
CL [mL/min/kg]	15.12	15.73 *	0.96	3.9%
**10 mg Oral in Adult Psychotic Otherwise Healthy** [48]				
AUC_inf [ng·h/mL]	212.8	200.5	1.06	6.1%
C_max [ng/mL]	9.13	8.34 *	1.08	9.5%
CL [mL/min/kg]	15	18.2	0.82	17.6%
**5 mg Oral in Adult Healthy** [49]				
AUC_inf [ng·h/mL]	40.10	44 *	0.91	8.9%
C_max [ng/mL]	1.28	1.27 *	1.01	0.8%
CL [mL/min/kg]	29.0	25.85 *	1.12	12.2%
**2 mg Oral in Adult Psychotic Otherwise Healthy** [46]				
AUC_inf [ng·h/mL]	90.73	103 *	0.88	12%
C_max [ng/mL]	3.95	3.54 *	1.12	11.6%
CL [mL/min/kg]	5.50	4.93 *	1.12	11.7%
**Predictability Assessment**				
	**AUC_inf**	**C_max**	**CL**	
**MFE**	0.99	1.03	0.99	
**RMSE**	27.27	1.66	1.96	

* Observed PK values were calculated using the PK-solver, otherwise, the reported values in the corresponding references were used.

**Table 6 pharmaceutics-14-01795-t006:** Comparison of PK parameters for haloperidol in the presence and absence of the CYP3A4 inducer rifampicin.

Group	Data	AUC__inf_ [ng·h/mL]	Half-Life [h]
**Control group**	Predicted	174	11.5
	Observed	186.1	10.1
	Fold error	0.93	1.22
**Rifampicin 600 mg**	Predicted	103.85	4.76
	Observed	105.03	4.93
	Fold error	0.99	0.97
**Pred. AUC ratio _rifampicin/control_**		0.60	
**Pred. T_1/2_ ratio _rifampicin/control_**			0.41

Ratio has been calculated based on prediction values.

## Data Availability

The data are available upon request.

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
