# Peer review of "Development and Evaluation of a Physiologically Based Pharmacokinetic Model for Predicting Haloperidol Exposure in Healthy and Disease Populations"

_pharmaceutics, 2022, doi:10.3390/pharmaceutics14091795_

Round 1

Reviewer 1 Report

This research article presents the results of a PBPK modeling and simulation exercise that incorporates data from published research studies on haloperidol in different patient populations, and aims to identify the most important variable(s) affecting haloperidiol PK. According to the authors, this is a novel undertaking as no covariates have been identified that would account for interindividual varibility in prior Pop-PK modeling efforts. The overall presentation and quality of the writing is excellent. jThis is a good benchmarking study of haloperidol PK and could be used as a reference for future studies that more carefully test the hypothesis presented hererin.

In terms of minor criticisms:

1)The authors' results suggest metabolism is the primary cause of interindividual variability in haloperidol PK. Nevertheless, there is no detailed explanation of the other factors that the authors considered when establishing that metabolism is the primary reason behind such variability. For example, could differences in plasma protein binding account for this variability? How about differences in body mass or BMI? Such comparisons would help the reader better gauge the importance of metabolism in relation to other variables. 

2) Why weren't variations in metabolism identified in Pop-PK studies as a source of interindividual variability? Such results should be apparent as variations in clearance/half-life -associated with the presence and absence of competitive inhibitors of metabolism -or not?

3) The grey, shaded areas in the curves need to be defined in the captions of figures 5 onward.

Author Response

Reviewers’ comments and the responses to their valuable comments

(Note: referees’ comments are in black, while the authors’ responses are in blue)

Reviewer #1 comments:

This research article presents the results of a PBPK modeling and simulation exercise that incorporates data from published research studies on haloperidol in different patient populations, and aims to identify the most important variable(s) affecting haloperidol PK. According to the authors, this is a novel undertaking as no covariates have been identified that would account for interindividual variability in prior Pop-PK modeling efforts. The overall presentation and quality of the writing is excellent. This is a good benchmarking study of haloperidol PK and could be used as a reference for future studies that more carefully test the hypothesis presented herein.

In terms of minor criticisms:

1)The authors' results suggest metabolism is the primary cause of interindividual variability in haloperidol PK. Nevertheless, there is no detailed explanation of the other factors that the authors considered when establishing that metabolism is the primary reason behind such variability. For example, could differences in plasma protein binding account for this variability? How about differences in body mass or BMI? Such comparisons would help the reader better gauge the importance of metabolism in relation to other variables.

Response: We are very grateful about your feeling regarding the overall presentation and quality of the writing. Actually, we agree that the sources of inter-individual variability are widely varied and can be related to many factors and can be explained in different ways. In comparison to what we suggest, a previous study evaluated the PK of haloperidol in terminally ill patients where several physiological factors (e.g. organ blood flows, tissue composition, and plasma proteins level) are usually altered [1]. The authors mentioned that they couldn’t explain the variability in haloperidol concentration based on the covariates that have been tested including body weights. They couldn’t find a strong and clear correlation in term of body weight and PK parameters. Moreover, body weights found not to be correlated with haloperidol clearance in a recently published Pop-PK analysis in terminally ill patients [2]. On the other hand, they found a negative correlation between C-reactive protein (CRP) and clearance of haloperidol. This can be used as an indicator that the inflammatory response may affect the biological exposure. However, a variation still exists in healthy volunteers, indicating that there are other possible mechanisms that such variations in plasma concentration might be attributed to. In Japanese study, they found that body weight was to somewhat extent correlated, but this was based on Japanese population [3], which might be attributed to the inherent differences related to ethnicity.

Generally, it is well documented that there is a positive correlation with body mass for the most of drugs [4]. However, in term of haloperidol, unexplained variation in plasma concentration was present even in a population with comparable baseline characteristics in term of ages and weights. Thus, the covariates could be considered as a source of the observed variation in the biological exposure of haloperidol, but we think that there are other factors that contribute significantly in such variation.

In term of protein binding, haloperidol is one of the drugs that are highly bound to plasma protein. Thus, its distribution is expected to be influenced, without regard to what extent. This was clearly estimated in our study as in case of TB patients where plasma proteins levels are usually affected. We were able to better estimate plasma profile and improve the predictability of the model in TB patients when we use plasma protein scaling factor to account for the effect of TB on plasma protein level.

We wanted to explain the unpredicted differences in plasma concentration of haloperidol in the normal individuals. Intrinsic enzymatic activities are widely varied between individuals [5]. Thus, we suggest that interindividual variability in haloperidol exposure can be mainly attributed to metabolism because that haloperidol is extensively metabolized in human. This property may cause a large interindividual variability in its kinetics behavior [5]. It has been mentioned that there is very wide range in intrinsic enzymatic activities in haloperidol metabolism (up to 10 fold). 

 (New paragraphs have been added summarizing these points. It can be easily detected in the manuscript since the track change function has been turned on)

2) Why weren't variations in metabolism identified in Pop-PK studies as a source of interindividual variability? Such results should be apparent as variations in clearance/half-life -associated with the presence and absence of competitive inhibitors of metabolism -or not?

Response: It is true and we completely agree. Pop-PK approach is very useful and can be used to find correlation between different covariates and PK parameters (e.g. clearance/half-life) of the drugs as the reviewer mentioned. Pop-PK analysis can be conducted with patient populations being stratified based on genetic polymorphism in enzymes and according to metabolizer phenotypes (poor metabolizer, intermediate, and rapid metabolizer, or ultra-rapid) or presence/absence of enzyme inhibitors. However, due to the involvement of several enzyme isoforms that are unequally contribute to haloperidol metabolism, it will be costly, and a large number of patients needed to be enrolled in the study. For example, in the Pop-PK analysis that was recently published [2], the involvement of glucuronidation and carbonyl reduction metabolic pathways were not considered in the stratification based on patient characteristics, even though that these metabolic pathways are accounted for more than 50 percent of haloperidol biotransformation.

The beauty of the PBPK approach is that we can model the metabolism with all enzymatic activities being incorporated, which will not be feasible in pop-PK. In contrast to Pop-PK where the model is fitted to the observed data (top-down approach), the PBPK enable us to predict the PK parameters from in vitro data even before conducting a clinical PK study (bottom-up approach).

3) The grey, shaded areas in the curves need to be defined in the captions of figures 5 onward. 

Response:  We thank the reviewer for pointing this out. We have defined the grey shaded areas in the captions in the revised version.

Reviewer 2 Report

In this study, Alasmari et al., developed a haloperidol PBPK model for identifying the potential sources of pharmacokinetic variability after intravenous and oral administration by using the population based simulator, PK-Sim. The model evaluation was performed using visual assessments, prediction error, and mean fold error of the ratio of the observed to predicted values of the PK-parameters. The predicted PK values were in good agreement with the corresponding reported values. This study was meaningful and should be considered for publication in Pharmaceutics without revision on my side.

Author Response

Reviewer #2 comments:

In this study, Alasmari et al., developed a haloperidol PBPK model for identifying the potential sources of pharmacokinetic variability after intravenous and oral administration by using the population-based simulator, PK-Sim. The model evaluation was performed using visual assessments, prediction error, and mean fold error of the ratio of the observed to predicted values of the PK-parameters. The predicted PK values were in good agreement with the corresponding reported values. This study was meaningful and should be considered for publication in Pharmaceutics without revision on my side

Response: We are extremely grateful for your opinion regarding our study to be considered for publication in in Pharmaceutics which is a highly reputable scientific journal. This will motivate our research’s group to move forward to advance the knowledge and contribute substantially to the pharmaceutical fields.

Reviewer 3 Report

The manuscript is well written and of high interest.  It is recommended that other drug-drug interactions where enzyme inhibition plays a role be included in future to test the PBPK model since these type of interactions are more commonly occurring than enzyme induction.    However, a good motivation was given why tuberculosis was chosen for drug-disease interaction as well as rifampicin (enzyme inducer) for drug-drug interaction on the pharmacokinetics of haloperidol, which is acceptable for development and evaluation of the model described in the current manuscript.

Author Response

Reviewer #3 comments:

The manuscript is well written and of high interest. It is recommended that other drug-drug interactions where enzyme inhibition plays a role be included in future to test the PBPK model since these type of interactions are more commonly occurring than enzyme induction.    However, a good motivation was given why tuberculosis was chosen for drug-disease interaction as well as rifampicin (enzyme inducer) for drug-drug interaction on the pharmacokinetics of haloperidol, which is acceptable for development and evaluation of the model described in the current manuscript.

Response: Our grateful thanks go to you for your kind words. Your recommendations for future studies are highly appreciated. We plan to utilize the developed model to evaluate other drug-interaction effect in the future. We believed that the PBPK model that we have developed in this study can be utilized in the future to evaluate the effect of enzyme inhibitors and guide clinical trials evaluating this effect with better use of the sources that are available.
